# A Sample Complexity Measure with Applications to Learning Optimal Auctions

**Vasilis Syrgkanis**
Microsoft Research
vasy@microsoft.com

## Abstract

We introduce a new sample complexity measure, which we refer to as split-sample growth rate. For any hypothesis $H$ and for any sample $S$ of size $m$, the split-sample growth rate $\hat{\tau}_H(m)$ counts how many different hypotheses can empirical risk minimization output on any sub-sample of $S$ of size $m/2$. We show that the expected generalization error is upper bounded by $O\left(\sqrt{\frac{\log(\hat{\tau}_H(2m))}{m}}\right)$. Our result is enabled by a strengthening of the Rademacher complexity analysis of the expected generalization error. We show that this sample complexity measure, greatly simplifies the analysis of the sample complexity of optimal auction design, for many auction classes studied in the literature. Their sample complexity can be derived solely by noticing that in these auction classes, ERM on any sample or sub-sample will pick parameters that are equal to one of the points in the sample.

## 1 Introduction

The seminal work of [11] gave a recipe for designing the revenue maximizing auction in auction settings where the private information of players is a single number and when the distribution over this number is completely known to the auctioneer. The latter raises the question of how has the auction designer formed this prior distribution over the private information. Recent work, starting from [4], addresses the question of how to design optimal auctions when having access only to samples of values from the bidders. We refer the reader to [5] for an overview of the existing results in the literature. [4, 9, 10, 2] give bounds on the sample complexity of optimal auctions without computational efficiency, while recent work has also focused on getting computationally efficient learning bounds [5, 13, 6].

This work solely focuses on sample complexity and not computational efficiency and thus is more related to [4, 9, 10, 2]. The latter work, uses tools from supervised learning, such as pseudo-dimension [12] (a variant of VC dimension for real-valued functions), compression bounds [8] and Rademacher complexity [12, 14] to bound the sample complexity of simple auction classes. Our work introduces a new measure of sample complexity, which is a strengthening the Rademacher complexity analysis and hence could also be of independent interest outside the scope of the sample complexity of optimal auctions. Moreover, for the case of auctions, this measure greatly simplifies the analysis of their sample complexity in many cases.

In particular, we show that in general PAC learning settings, the expected generalization error is upper bounded by the Rademacher complexity not of the whole class of hypotheses, but rather only over the class of hypotheses that could be the outcome of running Expected Risk Minimization (ERM) on a subset of the samples of half the size. If the number of these hypotheses is small, then the latter immediately yields a small generalization error. We refer to the growth rate of the latter set of hypotheses as the split-sample growth rate. This measure of complexity is not restricted to auction design and could be relevant to general statistical learning theory.

We then show that for many auction classes such as single-item auctions with player-specific reserves, single item $t$-level auctions and multiple-item item pricing auctions with additive buyers, the split-sample growth rate can be very easily bounded. The argument boils down to just saying that the Empirical Risk Minimization over this classes will set the parameters of the auctions to be equal to some value of some player in the sample. Then a simple counting argument gives bounds of the same order as in prior work in the literature that used the pseudo-dimension [9, 10]. In multi-item settings we also get improvements on the sample complexity bound.

Split-sample growth rate is similar in spirit to the notion of local Rademacher complexity [3], which looks at the Rademacher complexity on a subset of hypotheses with small empirical error. In particular, our proof is based on a refinement of the classic analysis Rademacher complexity analysis of generalization error (see e.g. [14]). However, our bound is more structural, restricting the set to outcomes of the chosen ERM process on a sub-sample of half the size. Moreover, we note that counting the number of possible outputs of ERM also has connections to a counting argument made in [1] in the context of pricing mechanisms. However, in essence the argument there is restricted to transductive settings where the sample "features" are known in advance and fixed and thereby the argument is much more straightforward and more similar to standard notions of "effective hypothesis space" used in VC-dimension arguments.

Our new measure of sample complexity is applicable in the general statistical learning theory framework and hence could have applications beyond auctions. To convey a high level intuition of settings where split-sample growth could simplify the sample complexity analysis, suppose that the output hypothesis of ERM is uniquely defined by a constant number of sample points (e.g. consider linear separators and assume that the loss is such that the output of ERM is uniquely characterized by choosing $O(d)$ points from the sample). Then this means that the number of possible hypotheses on any subset of size $m/2$, is at most $O(\binom{m}{d}) = O(m^d)$. Then the split sample growth rate analysis immediately yields that the expected generalization error is $O(\sqrt{d \cdot \log(m)/m})$, or equivalently the sample complexity of learning over this hypothesis class to within an $\epsilon$ error is $O(d \cdot \log(1/\epsilon)/\epsilon^2)$.

## 2 Preliminaries

We look at the sample complexity of optimal auctions. We consider the case of $m$ items, and $n$ bidders. Each bidder has a value function $v_i$ drawn independently from a distribution $D_i$ and we denote with $D$ the joint distribution.

We assume we are given a sample set $S = \{\mathbf{v}_1, \ldots, \mathbf{v}_m\}$, of $m$ valuation vectors, where each $\mathbf{v}_t \sim D$. Let $H$ denote the class of all dominant strategy truthful single item auctions (i.e. auctions where no player has incentive to report anything else other than his true value to the auction, independent of what other players do). Moreover, let

$$\mathbf{r}(h, \mathbf{v}) = \sum_{i=1}^{n} p_i^h(\mathbf{v}) \tag{1}$$

where $p_i^h(\cdot)$ is the payment function of mechanism $h$, and $\mathbf{r}(h, \mathbf{v})$ is the revenue of mechanism $h$ on valuation vector $\mathbf{v}$. Finally, let

$$\mathbf{R}_D(h) = \mathbb{E}_{\mathbf{v} \sim D}[\mathbf{r}(h, \mathbf{v})] \tag{2}$$

be the expected revenue of mechanism $h$ under the true distribution of values $D$.

Given a sample $S$ of size $m$, we want to compute a dominant strategy truthful mechanism $h_S$, such that:

$$\mathbb{E}_S[\mathbf{R}_D(h_S)] \geq \sup_{h \in H} \mathbf{R}_D(h) - \epsilon(m) \tag{3}$$

where $\epsilon(m) \to 0$ as $m \to \infty$. We refer to $\epsilon(m)$ as the *expected generalization error*. Moreover, we define the sample complexity of an auction class as:

**Definition 1** (Sample Complexity of Auction Class). *The (additive error) sample complexity of an auction class $H$ and a class of distributions $D$, for an accuracy target $\epsilon$ is defined as the smallest number of samples $m(\epsilon)$, such that for any $m \geq m(\epsilon)$:*

$$\mathbb{E}_S[\mathbf{R}_D(h_S)] \geq \sup_{h \in H} \mathbf{R}_D(h) - \epsilon \tag{4}$$

We might also be interested in a multiplcative error sample complexity, i.e.

$$\mathbb{E}_S\left[\mathrm{R}_D(h_S)\right] \geq (1 - \epsilon) \sup_{h \in H} \mathrm{R}_D(h) \tag{5}$$

The latter is exactly the notion that is used in [4, 5]. If one assumes that the optimal revenue on the distribution is lower bounded by some constant quantity, then an additive error implies a multiplicative error. For instance, if one assumes that player values are bounded away from zero with significant probability, then that implies a lower bound on revenue. Such assumptions for instance, are made in the work of [9]. We will focus on additive error in this work.

We will also be interested in proving high probability guarantees, i.e. with probability $1 - \delta$:

$$\mathrm{R}_D(h_S) \geq \sup_{h \in H} \mathrm{R}_D(h) - \epsilon(m, \delta) \tag{6}$$

where for any $\delta$, $\epsilon(m, \delta) \to 0$ as $m \to \infty$.

## 3 Generalization Error via the Split-Sample Growth Rate

We turn to the general PAC learning framework, and we give generalization guarantees in terms of a new notion of complexity of a hypothesis space $H$, which we denote as split-sample growth rate.

Consider an arbitrary hypothesis space $H$ and an arbitrary data space $Z$, and suppose we are given a set $S$ of $m$ samples $\{z_1, \ldots, z_m\}$, where each $z_t$ is drawn i.i.d. from some distribution $D$ on $Z$. We are interested in maximizing some reward function $\mathbf{r} : H \times Z \to [0, 1]$, in expectation over distribution $D$. In particular, denote with $\mathrm{R}_D(h) = \mathbb{E}_{z \sim D}\left[\mathbf{r}(h, z)\right]$.

We will look at the Expected Reward Maximization algorithm on $S$, with some fixed tie-breaking rule. Specifically, if we let

$$\mathrm{R}_S(h) = \frac{1}{m} \sum_{t=1}^{m} \mathbf{r}(h, z_t) \tag{7}$$

then ERM is defined as:

$$h_S = \arg \sup_{h \in H} \mathrm{R}_S(h) \tag{8}$$

where ties are broken based on some pre-defined manner.

We define the notion of a split-sample hypothesis space:

**Definition 2** (Split-Sample Hypothesis Space). *For any sample $S$, let $\hat{H}_S$, denote the set of all hypothesis $h_T$ output by the ERM algorithm (with the pre-defined tie-breaking rule), on any subset $T \subset S$, of size $\lceil |S|/2 \rceil$, i.e.:*

$$\hat{H}_S = \{h_T : T \subset S, |T| = \lceil |S|/2 \rceil\} \tag{9}$$

Based on the split-sample hypothesis space, we also define the split-sample growth rate of a hypothesis space $H$ at value $m$, as the largest possible size of $\hat{H}_S$ for any set $S$ of size $m$.

**Definition 3** (Split-Sample Growth Rate). *The split-sample growth rate of a hypothesis $H$ and an ERM process for $H$, is defined as:*

$$\hat{\tau}_H(m) = \sup_{S:|S|=m} |\hat{H}_S| \tag{10}$$

We first show that the generalization error is upper bounded by the Rademacher complexity evaluated on the split-sample hypothesis space of the union of two samples of size $m$. The Rademacher complexity $\mathcal{R}(S, H)$ of a sample $S$ of size $m$ and a hypothesis space $H$ is defined as:

$$\mathcal{R}(S, H) = \mathbb{E}_\sigma\left[\sup_{h \in H} \frac{2}{m} \sum_{z_t \in S} \sigma_t \cdot \mathbf{r}(h, z_t)\right] \tag{11}$$

where $\sigma = (\sigma_1, \ldots, \sigma_m)$ and each $\sigma_t$ is an independent binary random variable taking values $\{-1, 1\}$, each with equal probability.

**Lemma 1.** *For any hypothesis space $H$, and any fixed ERM process, we have:*

$$\mathbb{E}_S \left[ \mathrm{R}_D(h_S) \right] \geq \sup_{h \in H} \mathrm{R}_D(h) - \mathbb{E}_{S,S'} \left[ \mathcal{R}(S, \hat{H}_{S \cup S'}) \right], \tag{12}$$

*where $S$ and $S'$ are two independent samples of some size $m$.*

*Proof.* Let $h_*$ be the optimal hypothesis for distribution $D$. First we re-write the left hand side, by adding and subtracting the expected empirical reward:

$$
\begin{aligned}
\mathbb{E}_S \left[ \mathrm{R}_D(h_S) \right] &= \mathbb{E}_S \left[ \mathrm{R}_S(h_S) \right] - \mathbb{E}_S \left[ \mathrm{R}_S(h_S) - \mathrm{R}_D(h_S) \right] \\
&\geq \mathbb{E}_S \left[ \mathrm{R}_S(h_*) \right] - \mathbb{E}_S \left[ \mathrm{R}_S(h_S) - \mathrm{R}_D(h_S) \right] &&(h_S \text{ maximizes empirical reward}) \\
&= \mathrm{R}_D(h_*) - \mathbb{E}_S \left[ \mathrm{R}_S(h_S) - \mathrm{R}_D(h_S) \right] &&(h_* \text{ is independent of } S)
\end{aligned}
$$

Thus it suffices to upper bound the second quantity in the above equation.

Since $\mathrm{R}_D(h) = \mathbb{E}_{S'} \left[ \mathrm{R}_{S'}(h) \right]$ for a fresh sample $S'$ of size $m$, we have:

$$
\begin{aligned}
\mathbb{E}_S \left[ \mathrm{R}_S(h_S) - \mathrm{R}_D(h_S) \right] &= \mathbb{E}_S \left[ \mathrm{R}_S(h_S) - \mathbb{E}_{S'} \left[ \mathrm{R}_{S'}(h_S) \right] \right] \\
&= \mathbb{E}_{S,S'} \left[ \mathrm{R}_S(h_S) - \mathrm{R}_{S'}(h_S) \right]
\end{aligned}
$$

Now, consider the set $\hat{H}_{S \cup S'}$. Since $S$ is a subset of $S \cup S'$ of size $|S \cup S'|/2$, we have by the definition of the split-sample hypothesis space that $h_S \in \hat{H}_{S \cup S'}$. Thus we can upper bound the latter quantity by taking a supremum over $h \in \hat{H}_{S \cup S'}$:

$$
\begin{aligned}
\mathbb{E}_S \left[ \mathrm{R}_S(h_S) - \mathrm{R}_D(h_S) \right] &\leq \mathbb{E}_{S,S'} \left[ \sup_{h \in \hat{H}_{S \cup S'}} \mathrm{R}_S(h) - \mathrm{R}_{S'}(h) \right] \\
&= \mathbb{E}_{S,S'} \left[ \sup_{h \in \hat{H}_{S \cup S'}} \frac{1}{m} \sum_{t=1}^{m} \left( \mathbf{r}(h, z_t) - \mathbf{r}(h, z'_t) \right) \right]
\end{aligned}
$$

Now observe, that we can rename any sample $z_t \in S$ to $z'_t$ and sample $z'_t \in S'$ to $z_t$. By doing show we do not change the distribution. Moreover, we do not change the quantity $H_{S \cup S'}$, since $S \cup S'$ is invariant to such swaps. Finally, we only change the sign of the quantity $(\mathbf{r}(h, z_t) - \mathbf{r}(h, z'_t))$. Thus if we denote with $\sigma_t \in \{-1, 1\}$, a Rademacher variable, we get the above quantity is equal to:

$$\mathbb{E}_{S,S'} \left[ \sup_{h \in \hat{H}_{S \cup S'}} \frac{1}{m} \sum_{t=1}^{m} \left( \mathbf{r}(h, z_t) - \mathbf{r}(h, z'_t) \right) \right] = \mathbb{E}_{S,S'} \left[ \sup_{h \in \hat{H}_{S \cup S'}} \frac{1}{m} \sum_{t=1}^{m} \sigma_t \left( \mathbf{r}(h, z_t) - \mathbf{r}(h, z'_t) \right) \right] \tag{13}$$

for any vector $\sigma = (\sigma_1, \ldots, \sigma_m) \in \{-1, 1\}^m$. The latter also holds in expectation over $\sigma$, where $\sigma_t$ is randomly drawn between $\{-1, 1\}$ with equal probability. Hence:

$$\mathbb{E}_S \left[ \mathrm{R}_S(h_S) - \mathrm{R}_D(h_S) \right] \leq \mathbb{E}_{S,S',\sigma} \left[ \sup_{h \in \hat{H}_{S \cup S'}} \frac{1}{m} \sum_{t=1}^{m} \sigma_t \left( \mathbf{r}(h, z_t) - \mathbf{r}(h, z'_t) \right) \right]$$

By splitting the supremma into a positive and negative part and observing that the two expected quantities are identical, we get:

$$
\begin{aligned}
\mathbb{E}_S \left[ \mathrm{R}_S(h_S) - \mathrm{R}_D(h_S) \right] &\leq 2 \mathbb{E}_{S,S',\sigma} \left[ \sup_{h \in \hat{H}_{S \cup S'}} \frac{1}{m} \sum_{t=1}^{m} \sigma_t \mathbf{r}(h, z_t) \right] \\
&= \mathbb{E}_{S,S'} \left[ \mathcal{R}(S, \hat{H}_{S \cup S'}) \right]
\end{aligned}
$$

where $\mathcal{R}(S, H)$ denotes the Rademacher complexity of a sample $S$ and hypothesis $H$. ∎

Observe, that the latter theorem is a strengthening of the fact that the Rademacher complexity upper bounds the generalization error, simply because:

$$\mathbb{E}_{S,S'} \left[ \mathcal{R}(S, \hat{H}_{S \cup S'}) \right] \leq \mathbb{E}_{S,S'} \left[ \mathcal{R}(S, H) \right] = \mathbb{E}_S \left[ \mathcal{R}(S, H) \right] \tag{14}$$

Thus if we can bound the Rademacher complexity of $H$, then the latter lemma gives a bound on the generalization error. However, the reverse might not be true. Finally, we show our main theorem, which shows that if the split-sample hypothesis space has small size, then we immediately get a generalization bound, without the need to further analyze the Rademacher complexity of $H$.

**Theorem 2** (Main Theorem). *For any hypothesis space H, and any fixed ERM process, we have:*

$$\mathbb{E}_S\left[\mathbf{R}_D(h_S)\right] \geq \sup_{h \in H} \mathbf{R}_D(h) - \sqrt{\frac{2\log(\hat{\tau}_H(2m))}{m}} \qquad (15)$$

*Moreover, with probability $1 - \delta$:*

$$\mathbf{R}_D(h_S) \geq \sup_{h \in H} \mathbf{R}_D(h) - \frac{1}{\delta}\sqrt{\frac{2\log(\hat{\tau}_H(2m))}{m}} \qquad (16)$$

*Proof.* By applying Massart's lemma (see e.g. [14]) we have that:

$$\mathcal{R}(S, \hat{H}_{S \cup S'}) \leq \sqrt{\frac{2\log(|\hat{H}_{S \cup S'}|)}{m}} \leq \sqrt{\frac{2\log(\hat{\tau}_H(2m))}{m}} \qquad (17)$$

Combining the above with Lemma 1, yields the first part of the theorem.

Finally, the high probability statement follows from observing that the random variable $\sup_{h \in H} R_D(h) - R_D(h_S)$ is non-negative and by applying Markov's inequality: with probability $1 - \delta$

$$\sup_{h \in H} R_D(h) - R_D(h_S) \leq \frac{1}{\delta}\mathbb{E}_S\left[\sup_{h \in H} R_D(h) - R_D(h_S)\right] \leq \frac{1}{\delta}\sqrt{\frac{2\log(\hat{\tau}_H(2m))}{m}} \qquad (18)$$

∎

The latter theorem can be trivially extended to the case when $\mathbf{r} : H \times Z \to [\alpha, \beta]$, leading to a bound of the form:

$$\mathbb{E}_S\left[\mathbf{R}_D(h_S)\right] \geq \sup_{h \in H} \mathbf{R}_D(h) - (\beta - \alpha)\sqrt{\frac{2\log(\hat{\tau}_H(2m))}{m}} \qquad (19)$$

We note that unlike the standard Rademacher complexity, which is defined as $\mathcal{R}(S, H)$, our bound, which is based on bounding $\mathcal{R}(S, \hat{H}_{S \cup S'})$ for any two datasets $S, S'$ of equal size, does not imply a high probability bound via McDiarmid's inequality (see e.g. Chapter 26 of [14] of how this is done for Rademacher complexity analysis), but only via Markov's inequality. The latter yields a worse dependence on the confidence $\delta$ on the high probability bound of $1/\delta$, rather than $\log(1/\delta)$. The reason for the latter is that the quantity $\mathcal{R}(S, \hat{H}_{S \cup S'})$, depends on the sample $S$, not only in terms of on which points to evaluate the hypothesis, but also on determining the hypothesis space $\hat{H}_{S \cup S'}$. Hence, the function:

$$f(z_1, \ldots, z_m) = \mathbb{E}_{S'}\left[\sup_{h \in \hat{H}_{\{z_1,\ldots,z_m\} \cup S'}} \frac{1}{m}\sum_{t=1}^{m} \sigma_t\left(\mathbf{r}(h, z_t) - \mathbf{r}(h, z_t')\right)\right] \qquad (20)$$

does not satisfy the stability property that $|f(\mathbf{z}) - f(z_i'', \mathbf{z}_{-i})| \leq \frac{1}{m}$. The reason being that the supremum is taken over a different hypothesis space in the two inputs. This is unlike the case of the function:

$$f(z_1, \ldots, z_m) = \mathbb{E}_{S'}\left[\sup_{h \in H} \frac{1}{m}\sum_{t=1}^{m} \sigma_t\left(\mathbf{r}(h, z_t) - \mathbf{r}(h, z_t')\right)\right] \qquad (21)$$

which is used in the standard Rademacher complexity bound analysis, which satisfies the latter stability property. Resolving whether this worse dependence on $\delta$ is necessary is an interesting open question.

## 4   Sample Complexity of Auctions via Split-Sample Growth

We now present the application of the latter measure of complexity to the analysis of the sample complexity of revenue optimal auctions. Thoughout this section we assume that the revenue of any auction lies in the range $[0, 1]$. The results can be easily adapted to any other range $[\alpha, \beta]$, by

re-scaling the equations, which will lead to blow-ups in the sample complexity of the order of an extra $(\beta - \alpha)$ multiplicative factor. This limits the results here to bounded distributions of values. However, as was shown in [5], one can always cap the distribution of values up to some upper bound, for the case of regular distributions, by losing only an $\epsilon$ fraction of the revenue. So one can apply the results below on this capped distribution.

**Single bidder and single item.** Consider the case of a single bidder and single item auction. In this setting, it is known by results in auction theory [11] that an optimal auction belongs to the hypothesis class $H = \{\text{post a reserve price } r \text{ for } r \in [0,1]\}$. We consider, the ERM rule, which for any set $S$, in the case of ties, it favors reserve prices that are equal to some valuation $v_t \in S$. Wlog assume that samples $v_1, \ldots, v_m$ are ordered in increasing order. Observe, that for any set $S$, this ERM rule on any subset $T$ of $S$, will post a reserve price that is equal to some value $v_t \in T$. Any other reserve price in between two values $[v_t, v_{t+1}]$ is weakly dominated by posting $r = v_{t+1}$, as it does not change which samples are allocated and we can only increase revenue. Thus the space $\hat{H}_S$ is a subset of $\{\text{post a reserve price } r \in \{v_1, \ldots, v_m\}$. The latter is of size $m$. Thus the split-sample growth of $H$ is $\hat{\tau}_H(m) \leq m$. This yields:

$$\mathbb{E}_S\left[\mathrm{R}_D(h_S)\right] \geq \sup_{h \in H} \mathrm{R}_D(h) - \sqrt{\frac{2\log(2m)}{m}} \tag{22}$$

Equivalently, the sample complexity is $m_H(\epsilon) = O\left(\frac{\log(1/\epsilon)}{\epsilon^2}\right)$.

**Multiple i.i.d. regular bidders and single item.** In this case, it is known by results in auction theory [11] that the optimal auction belongs to the space of hypotheses $H$ consisting of second price auctions with some reserve $r \in [0,1]$. Again if we consider ERM which in case of ties favors a reserve that equals to a value in the sample (assuming that is part of the tied set, or outputs any other value otherwise), then observe that for any subset $T$ of a sample $S$, ERM on that subset will pick a reserve price that is equal to one of the values in the samples $S$. Thus $\hat{\tau}_H(m) \leq n \cdot m$. This yields:

$$\mathbb{E}_S\left[\mathrm{R}_D(h_S)\right] \geq \sup_{h \in H} \mathrm{R}_D(h) - \sqrt{\frac{2\log(2 \cdot n \cdot m)}{m}} \tag{23}$$

Equivalently, the sample complexity is $m_H(\epsilon) = O\left(\frac{\log(n/\epsilon^2)}{\epsilon^2}\right)$.

**Non-i.i.d. regular bidders, single item, second price with player specific reserves.** In this case, it is known by results in auction theory [11] that the optimal auction belongs to the space of hypotheses $H_{SP}$ consisting of second price auctions with some reserve $r_i \in [0,1]$ for each player $i$. Again if we consider ERM which in case of ties favors a reserve that equals to a value in the sample (assuming that is part of the tied set, or outputs any other value otherwise), then observe that for any subset $T$ of a sample $S$, ERM on that subset will pick a reserve price $r_i$ that is equal to one of the values $v_t^i$ of player $i$ in the sample $S$. There are $m$ such possible choices for each player, thus $m^n$ possible choices of reserves in total. Thus $\hat{\tau}_H(m) \leq m^n$. This yields:

$$\mathbb{E}_S\left[\mathrm{R}_D(h_S)\right] \geq \sup_{h \in H_{SP}} \mathrm{R}_D(h) - \sqrt{\frac{2n\log(2m)}{m}} \tag{24}$$

If $H$ is the space of all dominant strategy truthful mechanisms, then by prophet inequalities (see [7]), we know that $\sup_{h \in H_{SP}} \mathrm{R}_D(h) \geq \frac{1}{2} \sup_{h \in H} \mathrm{R}_D(h)$. Thus:

$$\mathbb{E}_S\left[\mathrm{R}_D(h_S)\right] \geq \frac{1}{2} \sup_{h \in H} \mathrm{R}_D(h) - \sqrt{\frac{2n\log(2m)}{m}} \tag{25}$$

**Non-i.i.d. irregular bidders single item.** In this case it is known by results in auction theory [11] that the optimal auction belongs to the space of hypotheses $H$ consisting of all virtual welfare maximizing auctions: For each player $i$, pick a monotone function $\hat{\phi}_i(v_i) \in [-1,1]$ and allocate to the player with the highest non-negative virtual value, charging him the lowest value he could have bid and still win the item. In this case, we will first coarsen the space of all possible auctions.

In particular, we will consider the class of $t$-level auctions of [9]. In this class, we constrain the value functions $\hat{\phi}_i(v_i)$ to only take values in the discrete $\epsilon$ grid in $[0, 1]$. We will call this class $H_\epsilon$. An equivalent representation of these auctions is by saying that for each player $i$, we define a vector of thresholds $0 = \theta_0^i \leq \theta_1^i \leq \ldots \leq \theta_s^i \leq \theta_{s+1}^i = 1$, with $s = 1/\epsilon$. The index of a player is the largest $j$ for which $v_i \geq \theta_j$. Then we allocate the item to the player with the highest index (breaking ties lexicographically) and charge the minimum value he has to bid to continue to win.

Observe that on any sample $S$ of valuation vectors, it is always weakly better to place the thresholds $\theta_j^i$ on one of the values in the set $S$. Any other threshold is weakly dominated, as it does not change the allocation. Thus for any subset $T$ of a set $S$ of size $m$, we have that the thresholds of each player $i$ will take one of the values of player $i$ that appears in set $S$. We have $1/\epsilon$ thresholds for each player, hence $m^{1/\epsilon}$ combinations of thresholds for each player and $m^{n/\epsilon}$ combinations of thresholds for all players. Thus $\hat{\tau}_H(m) \leq m^{n/\epsilon}$. This yields:

$$\mathbb{E}_S\left[\mathbf{R}_D(h_S)\right] \geq \sup_{h \in H_\epsilon} \mathbf{R}_D(h) - \sqrt{\frac{2n \log(2m)}{\epsilon \cdot m}} \tag{26}$$

Moreover, by [9] we also have that:

$$\sup_{h \in H_\epsilon} \mathbf{R}_D(h) \geq \sup_{h \in H} \mathbf{R}_D(h) - \epsilon \tag{27}$$

Picking, $\epsilon = \left(\frac{2n \log(2m)}{m}\right)^{1/3}$, we get:

$$\mathbb{E}_S\left[\mathbf{R}_D(h_S)\right] \geq \sup_{h \in H} \mathbf{R}_D(h) - 2\left(\frac{2n \log(2m)}{m}\right)^{1/3} \tag{28}$$

Equivalently, the sample complexity is $m_H(\epsilon) = O\left(\frac{n \log(1/\epsilon)}{\epsilon^3}\right)$.

**$k$ items, $n$ bidders, additive valuations, grand bundle pricing.** If the reserve price was anonymous, then the reserve price output by ERM on any subset of a sample $S$ of size $m$, will take the value of one of the $m$ total values for the items of the buyers in $S$. So $\hat{\tau}_H(m) = m \cdot n$. If the reserve price was not anonymous, then for each buyer ERM will pick one of the $m$ total item values, so $\hat{\tau}_H(m) \leq m^n$. Thus the sample complexity is $m_H(\epsilon) = O\left(\frac{n \log(1/\epsilon)}{\epsilon^2}\right)$.

**$k$ items, $n$ bidders, additive valuations, item prices.** If reserve prices are anonymous, then each reserve price on item $j$ computed by ERM on any subset of a sample $S$ of size $m$, will take the value of one of the player's values for item $j$, i.e. $n \cdot m$. So $\hat{\tau}_H(m) = (n \cdot m)^k$. If reserve prices are not anonymous, then the reserve price on item $j$ for player $i$ will take the value of one of the player's values for the item. So $\hat{\tau}_H(m) \leq m^{n \cdot k}$. Thus the sample complexity is $m_H(\epsilon) = O\left(\frac{nk \log(1/\epsilon)}{\epsilon^2}\right)$.

**$k$ items, $n$ bidders, additive valuations, best of grand bundle pricing and item pricing.** ERM on the combination will take values on any subset of a sample $S$ of size $m$, that is at most the product of the values of each of the classes (bundle or item pricing). Thus, for anonymous pricing: $\hat{\tau}_H(m) = (m \cdot n)^{k+1}$ and for non-anonymous pricing: $\hat{\tau}_H(m) \leq m^{n(k+1)}$. Thus the sample complexity is $m_H(\epsilon) = O\left(\frac{n(k+1) \log(1/\epsilon)}{\epsilon^2}\right)$.

In the case of a single bidder, we know that the best of bundle pricing or item pricing is a $1/8$ approximation to the overall best truthful mechanism for the true distribution of values, assuming values for each item are drawn independently. Thus in the latter case we have:

$$\mathbb{E}_S\left[\mathbf{R}_D(h_S)\right] \geq \frac{1}{6} \sup_{h \in H} \mathbf{R}_D(h) - \sqrt{\frac{2(k+1) \log(2m)}{m}} \tag{29}$$

where $H$ is the class of all truthful mechanisms.

**Comparison with [10].** The latter three applications were analyzed by [10], via the notion of the pseudo-dimension, but their results lead to sample complexity bounds of $O(\frac{nk \log(nk) \log(1/\epsilon)}{\epsilon^2})$. Thus the above simpler analysis removes the extra log factor on the dependence.

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
