[Reviews · NeurIPS 2017]

Reviewer 1



Recently there has been a lot of interest in the Algorithmic Game Theory community in designing approximately revenue optimal auctions given a small number of samples of the buyer's valuations. A pertinent question is what is the minimum number of samples required to design a truthful auction that provides an additive epsilon approximation to the optimal revenue. A bound on this quantity is usually obtained by relating it to learning theory concepts like VC dimension and pseudo-dimension. This paper proposes a new notion of split-sample complexity and applies to obtain bound on the sample complexity for common auction scenarios. The notion of split-sample complexity is as follows: Given a set of m samples, for any given subset of size m/2, we can identify an optimal hypothesis. Denote by \hat{H}_S the set of optimal hypotheses over all size m/2 subsets of S. The split sample growth rate is how large the set \hat{H}_S can get over all sets of size S. The authors show that given m samples, the expected revenue obtained from the optimal hypothesis over these m samples, can be related to the optimal revenue for the given distribution and an additive error term. This additive error term relates to the split-sample complexity as a function of m. Then for specific classes of auctions sample complexity bounds can be obtained by bounding the split-sample complexity. This new approach allows the author to strengthen previous bounds for bundle and item pricing mechanisms by Morgernstern and Roughgarden. Overall this paper is well written and provides a new technique for tackling an ongoing research agenda.

Reviewer 2



The paper introduces a new hypothesis class complexity measure called "split-sample growth rate", which counts the number of possible hypotheses ERM can output on a subsample of half the data, and proves generalization bounds using this measure. The utility of this measure is demonstrated by application to optimal auctions, which have particularly simple split-sample growth rates. I found the simplicity of the approach compelling, and the results significant, though the paper was quite dense and not easy to read in places. A major omission in my opinion is a conclusion which comments on the potential for "split-sample growth rate" to find applications beyond auctions, and I ask the authors to comment on this in the author response. Line 23- remove first comma Line 106- by doing so- not show Lemma 1- statement and proof are fairly dense and difficult to understand Line 160- missing closing } Line 162- it would be nice if the authors could show this derivation

Reviewer 3



Summary: Rademacher complexity is a powerful tool for producing generalization guarantees. The Rademacher complexity of a class H on a sample S is roughly the expected maximum gap between training and test performance if S were randomly partitioned into training and test sets, and we took the maximum over all h in H (eqn 11). This paper makes the observation that the core steps in the standard generalization bound proof will still go through if instead of taking the max over all h in H, you only look at the h's that your procedure can possibly output when given half of a double sample (Lemma 1). While unfortunately the usual final (or initial) high-probability step in this argument does not seem to go through directly, the paper shows (Theorem 2) that one can nonetheless get a useful generation bound from this using other means. The paper then shows how this generalization bound yields good sample complexity guarantees for a number of natural auction classes. In several cases, this improves over the best prior guarantees known, and also simplifies their analysis. Evaluation: While the core idea is to some extent an observation, I like the paper because it is a nice, cute idea that one could actually teach in a class. And it allows for a simpler sample complexity analysis for auctions, which is great. It may also have other uses too. On the negative side, I wish the paper were able to answer whether the linear dependence on 1/delta is really necessary (see below). That would make the paper feel more complete to me. Still, I am overall positive. Questions: Do you know if the linear dependence on delta in Theorem 2 is required? I see why you need it in your proof (due to using Markov) and you explain nicely why the usual use of McDiarmid to get an O(log(1/delta)) bound doesn't go through. But this leaves open whether (a) there is a different way to get an O(log(1/delta)) bound, or on the other hand (b) there exist D,H for which you indeed have a more heavy-tailed chance of failure. If this were resolved, the paper would feel more complete. On a related note, do you know if you can get a bound along the lines of Theorem 2 in terms of the maximum # of hypotheses that can be produced from half of the *actual* sample S? (That is, replacing the sup over S of size m in the definition of \hat{\tau} with the actual training sample S?) Suggestions: - If I am not mistaken, The proof of Lemma 1 basically goes through the textbook Rademacher bound proof, changing it where needed to replace H with \hat{H}_{S \union S'}. That's fine, but you should explicitly tell the reader that that is what you are doing up front, so they know what's new vs what's old. E.g., you have a nice discussion of this form for Theorem 2, where you talk about where it differs from the usual argument, that I think is quite useful. - In Section 4 you examine several interesting scenarios (single bidder, multiple iid regular,...) , and for each one you say "In this case, the space of hypotheses H is ...". I think what you mean to say is more that "In this case, it is known that an *optimal auction belongs to* the hypothesis class H of ...". In other words, the generalization guarantees apply to any H you want, but now you are saying what the meaningful H is in each scenario, right? I think this is worth saying explicitly, since otherwise the reader might get confused why H is changing each time. Also I'll point out that the idea of counting the number of possible outputs of an auction mechanism in order to get a sample complexity bound has been used in the auction literature before. The first I know of is the FOCS 2005 paper "Mechanism design via machine learning" by Balcan, Blum, Hartline, and Mansour, though that paper operates in a transductive setting where an argument of this type is much easier to make.